

# Machine learning-based prediction of LDL cholesterol: performance evaluation and validation

Jing-Bi Meng[1,*], Zai-Jian An[2,*] and Chun-Shan Jiang[2]

[1] Central Laboratory, Yanbian University Hospital, Yanji, Jilin Province, China
[2] Department of Clinical Laboratory, Yanbian University Hospital, Yanji, Jilin Province, China
[*] These authors contributed equally to this work.

## ABSTRACT

**Objective**. This study aimed to validate and optimize a machine learning algorithm for accurately predicting low-density lipoprotein cholesterol (LDL-C) levels, addressing limitations of traditional formulas, particularly in hypertriglyceridemia.

**Methods**. Various machine learning models—linear regression, K-nearest neighbors (KNN), decision tree, random forest, eXtreme Gradient Boosting (XGB), and multilayer perceptron (MLP) regressor—were compared to conventional formulas (Friedewald, Martin, and Sampson) using lipid profiles from 120,174 subjects (2020–2023). Predictive performance was evaluated using R-squared ($R^2$), mean squared error (MSE), and Pearson correlation coefficient (PCC) against measured LDL-C values.

**Results**. Machine learning models outperformed traditional methods, with Random Forest and XGB achieving the highest accuracy ($R^2 = 0.94$, MSE = 89.25) on the internal dataset. Among the traditional formulas, the Sampson method performed best but showed reduced accuracy in high triglyceride (TG) groups (TG > 300 mg/dL). Machine learning models maintained high predictive power across all TG levels.

**Conclusion**. Machine learning models offer more accurate LDL-C estimates, especially in high TG contexts where traditional formulas are less reliable. These models could enhance cardiovascular risk assessment by providing more precise LDL-C estimates, potentially leading to more informed treatment decisions and improved patient outcomes.

# INTRODUCTION

Low-density lipoprotein cholesterol (LDL-C) represents the final product of lipoprotein metabolism and is linked to an increased risk of cardiovascular disease (CVD) mortality (*Zhou et al., 2020*). Notably, the accumulation and oxidation of LDL-C within the arterial intima constitute a significant modifiable risk factor for atherosclerotic cardiovascular diseases (ASCVD) (*Malekmohammad, Bezsonov & Rafieian-Kopaei, 2021*). The National Cholesterol Education Program (NCEP) Adult Treatment Panel III (ATP III) designates LDL-C as the primary biomarker for assessing ASCVD risk and the central target for lipid-lowering therapy (*Expert Panel on Detection E, and Treatment of High Blood Cholesterol in*

Corresponding author
Chun-Shan Jiang, jcs0915@163.com

*Adults, 2001*), which underscores the pivotal role assigned to LDL-C in assessing ASCVD risk and guiding interventions to lower lipid levels.

In Chinese community residents, 33.8% of individuals experience lipid abnormalities, a phenomenon that has emerged as a significant public health issue in China (*Lu et al., 2021*). On the other hand, a portion of Chinese community hospitals is subjected to limitations in obtaining accurate LDL-C measurements due to cost constraints, which could potentially exacerbate the prevalence of lipid abnormalities. Therefore, ensuring the precision and reliability of LDL-C determination is crucial in diagnostic cardiology, thereby presenting specific challenges for clinical laboratories and emphasizing the fundamental importance of accurate LDL-C assessment.

LDL-C is routinely calculated using the Friedewald equation, which incorporates data from the standard lipid panel (total cholesterol (TC), high-density lipoprotein cholesterol (HDL-C), and triglycerides (TG)): LDL-C = TC − (HDL-C) − (TG/5) (*Friedewald, Levy & Fredrickson, 1972*). The accuracy of the equation for high-TG samples is compromised by the variability in the ratio of cholesterol to TG in LDL-C and influenced by factors such as TG size and other considerations (*Friedewald, Levy & Fredrickson, 1972*). The Friedewald equation is not applicable in scenarios characterized by severe hypertriglyceridemia, herein as TG >400 mg/dL (4.52 mmol/L), very low LDL-C (LDL-C <70 mg/dL [1.81 mmol/L]), and diabetes (*Ferrinho et al., 2021*; *Palmer et al., 2019*). In such scenarios, the Friedewald equation tends to markedly underestimate LDL-C. In 2013, *Martin et al. (2013b)* developed the Martin equation for LDL-C estimation as a solution for addressing these inaccuracies. The equation involves subtracting the HDL-C and TG/adjustable factor from TC, with the adjustable factor representing the strata-specific median TG: very low density lipoprotein cholesterol (VLDL-C) ratios. The Martin equation, which is developed using traditional linear regression analysis and exhibits superior accuracy compared to the Friedewald formula, still retains inaccuracies, particularly in the context of lower LDL-C estimates (*Quispe et al., 2017*). In 2020, the Sampson equation emerged as a novel development. Like the Martin equation, it exhibits superior accuracy over the Friedewald equation, especially in scenarios involving patients with hypertriglyceridemia (*Sampson et al., 2020*; *Sampson et al., 2022*). Unlike the Martin equation, the Sampson equation was formulated using the beta quantification (BQ) reference method, which incorporates a swinging bucket ultracentrifuge procedure along with an additional step for LDL precipitation.

Recently, a variety of machine learning (ML) methods have been applied in clinical laboratory-associated research (*Kurstjens et al., 2022*; *Rangan et al., 2022*). In the era of precision medicine, to further enhance the estimation of LDL-C, we employed various methods based on ML regression algorithms (*Chan & Veas, 2024*; *Pedregosa et al., 2011*), which were utilized to derive an optimal approach for estimating LDL-C from the standard lipid panel. Subsequently, to underscore the accuracy advantages of the latest algorithm, we compared it with direct measurements of LDL-C, as well as the Friedewald, Martin, and Sampson LDL-C estimation methods.

## MATERIALS AND METHODS

### Study population

This was a retrospective study, all data used were anonymized and there was no direct contact or intervention with the study subjects, and it was approved by the Ethics Review Committee of Yanbian University Hospital, Ethics No. 2024665. The cohort comprised samples of consecutive standard lipid profile samples, including directly measured components of TC, HDL-C, and TG, alongside the corresponding directly measured LDL-C values. Collected between January 1, 2020, and March 31, 2023, at Yanbian University Hospital's inpatient and outpatient units for clinical indications, the inclusion criteria stipulated determining directly measured components of a standard lipid profile (TC, TG, HDL-C) and directly measured LDL-C on the same day; thus, day-to-day variations in cholesterol particles were minimized. Data extraction was conducted through the laboratory information system (LIS) system. All continuous variables in this study underwent Kolmogorov–Smirnov testing (*Srimani et al., 2021*). A significance level of $P > 0.05$ was utilized to assess the conformity of the data to the assumption of normal distribution.

### Lipid profile testing

The serum levels of TG, TC, HDL-C, and LDL-C were measured in the clinical laboratory of Yanbian University using the Roche Cobas702 chemistry analyzer. The analyzer undergoes calibration every 14 days, and quality control measures adhere to the regulations and certification requirements established by the Jilin Provincial Government.

We utilized the cholesterol oxidase-peroxidase-aminoantipyrine phenol (CHOD-PAP) method to estimate TC and the glycerol phosphate oxidase-peroxidase-aminoantipyrine phenol (GPO-PAP) enzymatic colorimetric method to estimate TG (*Rifai, 2006*). The detection method for LDL-C is the surfactant LDL-C assay, and for HDL-C, the catalase HDL-C assay is utilized. The measurements demonstrate linearity, with the TG range being 44.3–1,000 mg/dL (0.5–11.3 mmol/L), the TC range being 19.3–500 mg/dL (0.5–12.9 mmol/L), the HDL-C range being 3.8–96.7 mg/dL (0.1–2.5 mmol/L), and the LDL-C range being 7.7–450 mg/dL (0.2–11.6 mmol/L). Calibration for TC and TG is performed every 15 days, whereas LDL-C and HDL-C are calibrated daily. Two levels of quality control materials (high and low limits) are analyzed each day.

### Data preprocessing

The first screening step involved excluding patients with missing values in TC, TG, HDL-C, LDL-C, age, and gender and patients with values outside the detection range were, subsequently, excluded due to the relative deviation of outlier values exceeding 10%. Data from January 2020 to December 2022 was utilized as the internal training and validation set, whereas data from January to March 2023 served as the secondary internal validation dataset. Within the internal dataset, patient data points with identical features but different actual values were removed. Then, we employed the multi-model consensus approach involving RandomForestRegressor, DecisionTreeRegressor, and XGBRegressor to determine feature importances (*Pedregosa et al., 2011*). Each model was trained on

 

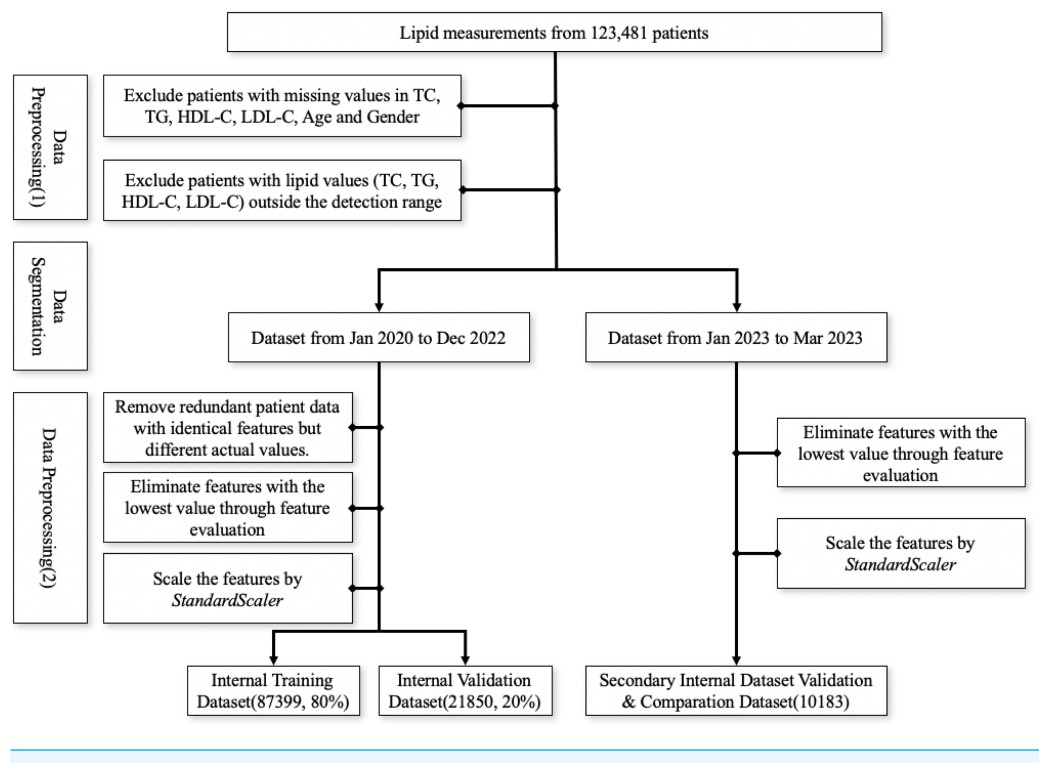

**Figure 1  Workflow for data preprocessing.**

the same dataset, and their feature importances were averaged. Based on the averaged feature importances, features identified as minimally important were removed (Fig. 1). Consequently, we used the StandardScaler from the sklearn.preprocessing package for feature scaling. First, we initialized the StandardScaler object and applied the fit operation on the training set. When scaling the features of the internal validation dataset, we utilized the previously saved StandardScaler object and applied the transform operation to ensure that the scaling parameters derived from the training set were used during the parameter calculation.

## Machine learning algorithm and assessment methods

Using the Scikit-learn application programming interface (API) (*Pedregosa et al., 2011*), we conducted ML analysis. To predict LDL-C values based on the actual measurements of TC, TG, and HDL-C, we constructed various models including linear, K-nearest neighbors (KNN), decision trees, random forest, eXtreme Gradient Boost (XGBoost), and multi-layer perceptron (MLP) regression models. The directly measured LDL-C was utilized as the truth label.

We divided the dataset into training (80%) and test sets (20%). To optimize hyperparameters, we used a combination of grid search and 5-fold cross-validation across various settings. The hyperparameters we explored included learning rate, tree depth, minimum samples per leaf, number of estimators for ensemble methods, and layers/neurons configuration for neural networks. The tuning process aimed to find

**Table 1  Equations for LDL-C estimation.**

| | |
|---|---|
| Friedewald Equations | $LDL\_C(mmol/L) = TC - HDL\_C - TG/5$ |
| Martin Equations | $LDL\_C(mg/dl) = TC - HDL\_C - TG/X(adjustable\ coefficient)$ |
| Sampson Equations | $LDL\_C(mg/dl) = TC/0.948 - HDL\_C/0.971 - [TG/8.59 + (TG - Non\_HDL\_C)/2140 - TG*TG/16100] - 9.44$ |

**Table 2  Conversions between mmol/L and mg/dL.**

| mmol/L → mg/dL |
|---|
| TG (mmol/L) * 88.57 = TG (mg/dL) |
| TC (mmol/L) * 38.67 = TC (mg/dL) |
| HDL-C (mmol/L) * 38.67 = HDL-C (mg/dL) |
| LDL-C (mmol/L) * 38.67 = LDL-C (mg/dL) |

the best hyperparameter combination that minimized mean squared error (MSE) while maximizing $R^2$ and Pearson correlation coefficient (PCC) on the validation set, ensuring model generalizability.

Following the identification of the optimal ML model through internal validation, the model was tested on a secondary internal validation set. The dataset used for this secondary internal validation was split by time, with data from January 2023 to March 2023 serving as the validation set. This approach ensures that the model is tested on data collected at a later point in time, though it is still derived from the same source as the training data.

Additionally, we compared the performance of the model against predictions from the Friedewald formula, Sampson formula, and Martin equation on the same validation set.

## LDL calculation formulas

LDL-C was calculated using the Friedewald, Martin, and Sampson formulas (Table 1). For the Martin equation, the LDL calculator available at (http://www.LDLCalculator.com) was employed to determine LDL-C values, whereas the remaining calculations were performed using Microsoft Excel 2021. Additionally, conversions between mmol/L and mg/dL were referenced from Table 2.

## Statistics analysis

To evaluate the accuracy of the ML models and existing LDL-C formulas, we utilized three common metrics: $R^2$, MSE, and PCC. $R^2$ measures the proportion of the variance in the dependent variable that is predictable from the independent variables. An $R^2$ value closer to 1 indicates a higher predictive accuracy, whereas an $R^2$ value below 0 implies that the model performs worse than simply predicting the mean (*Chicco, Warrens & Jurman, 2021*). This can occur when the model fails to capture the relationship between the variables, particularly in cases with extreme outliers or when the data does not fit well to the model structure.

Lower MSE values, which indicate more optimal model performance, reflect the average squared difference between the predicted and actual values. The Pearson correlation

coefficient (PCC) assesses the linear relationship between two variables, with values closer to 1 indicating a strong positive correlation. Models or formulas with higher $R^2$ values, lower MSE values, and higher Pearson correlation coefficients were considered to exhibit superior predictive accuracy. All statistical analyses were performed using Python version 3.11.5.

Due to the significant variability that affects the performance of existing formulas across different triglyceride (TG) ranges, we categorized the test data into six groups based on TG levels: TG <100 mg/dL, 100 mg/dL ≤ TG <150 mg/dL, 150 mg/dL ≤ TG <200 mg/dL, 200 mg/dL ≤ TG <300 mg/dL, 300 mg/dL ≤ TG <400 mg/dL, and TG ≥ 400 mg/dL. Subsequently, we evaluated the performance of the ML models and that of the existing formulas within these defined TG groups. This systematic approach facilitated a comprehensive assessment of model performance across the spectrum of triglyceride levels, thus enhancing the robustness and applicability of the findings.

## RESULTS

### Original clinical data

We present the baseline characteristics of the study subjects in each dataset (Table 3). All continuous variables failed the normality test. Between January 1, 2020, and March 31, 2023, we conducted a comprehensive lipid profile study encompassing 120,174 unique individuals (63,392 males; 52.8%), ranging from 1 to 103 years. All datasets exhibited a higher proportion of male subjects than female ones. In the internal dataset, the LDL-C median value was higher than that in the secondary internal validation dataset, whereas the values of TC and TG in the internal dataset were lower than those in the secondary internal validation dataset. As part of the internal training and testing dataset, a total of 109,991 cases, spanning from January 1, 2020, to December 31, 2022, were included. In this subgroup, the median LDL-C level was 116.78 mg/dL, TC level was 178.66 mg/dL, and TG level was 135.51 mg/dL. For the secondary internal validation dataset covering patients from January 1, 2023, to March 31, 2023, a total of 10,183 cases were evaluated. In this distinct cohort, the median LDL-C level was 110.21 mg/dL, TC level was 184.46 mg/dL, and TG level was 138.17 mg/dL (Table 3).

### Feature importances

The feature-weight graph was generated using a multi-model consensus approach involving RandomForestRegressor, DecisionTreeRegressor, and XGBRegressor. Each model was trained on the same dataset, and their feature importances were averaged. TC exhibited the highest importance, while gender had the lowest (Fig. 2, Table S1). To optimize computational resources, the gender feature, with the lowest weight importance, was consequently excluded from further analysis.

The feature weight graph visually represents the relative importance of each feature in the predictive model. Features with higher weights contribute more significantly to the model's predictions, whereas those with lower weights exert less impact. In this context, TC emerges as the most influential feature, thereby indicating its strong association with the target variable.

**Table 3  Baseline characteristics of the study subjects.**

| | | Inner dataset | Secondary internal validation dataset |
|---|---|---|---|
| | | *n* = 109,249 | *n* = 10,183 |
| Age | | 57 (46,66) | 58 (49,67) |
| Gender | | | |
| | Male | 57,774 (52.88%) | 5,208 (51.14%) |
| | Female | 51,475 (47.12%) | 4,975 (48.86%) |
| Lipid Profile | | | |
| | TC | 178.66 (143.85, 208.04) | 184.46 (148.88, 214.23) |
| | TG | 135.51 (93.88, 200.17) | 138.17 (97.43, 201.05) |
| | HDL-C | 43.31 (35.58, 52.20) | 41.74 (35.19, 49.50) |
| | LDL-C | 116.78 (85.07, 141.53) | 110.21 (80.05, 136.89) |

**Notes.**

Data are expressed as medians (interquartile range) for continuous variables and frequencies (percentages) for categorical variables.

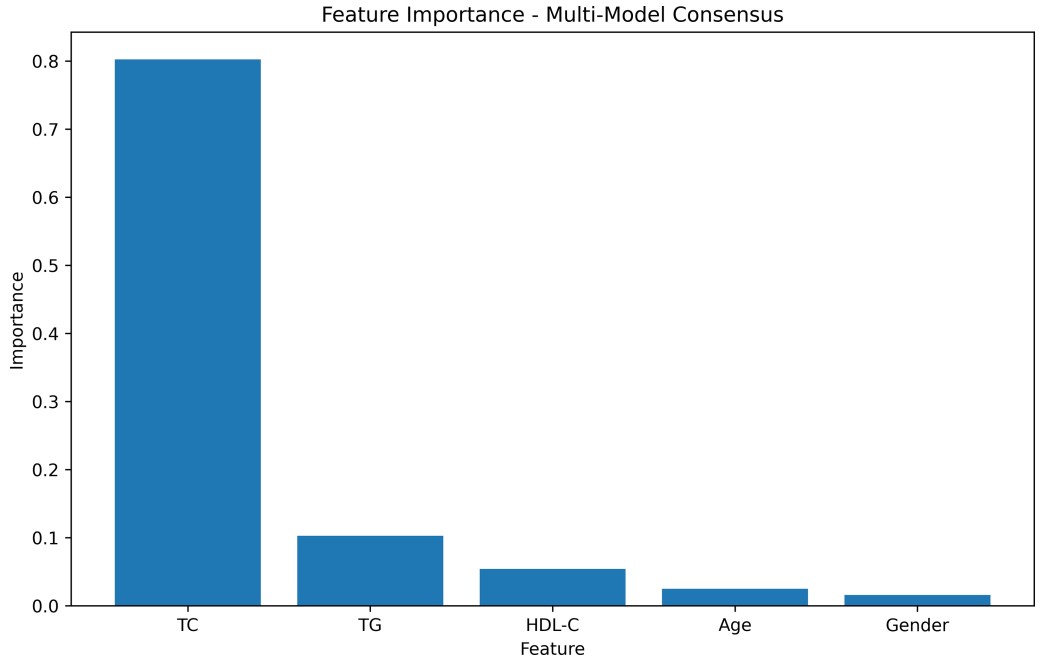

**Figure 2  Feature importances.**

## Inner training and test

In the learning curve plots, it is evident that none of the models exhibit significant underfitting. Although slight overfitting is observed in the decision tree and random forest models, it does not affect the model training. From the scatter plots, the distribution of actual values and model predicted values, along with the differences between residual and predicted values, are examined to assess model fitting. The decision tree model exhibits the lowest scores ($R^2 = 0.843$, PCC=0.918, MSE=257.522), whereas ensemble learning

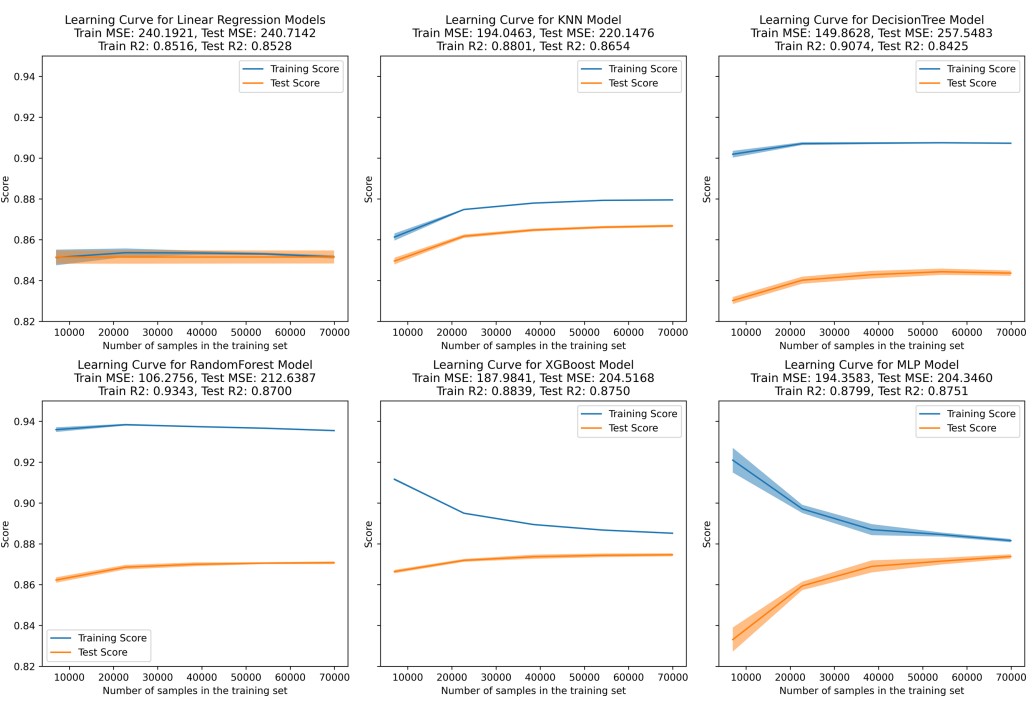

**Figure 3** **Learning curve for machine learning model.** (A–E) represent individual models, with blue lines denoting internal training scores and orange lines representing internal test scores on the $R^2$ scale.

methods (random forest and XGBoost) and the neural network model (MLP) demonstrate higher scores compared to other models (Figs. 3–5). To further understand the contribution of each feature to the model's training, we conducted an analysis of feature importance (Fig. S1). The results reveal that total cholesterol (TC) is the most influential feature across both the decision tree and XGBoost models, with importance scores of 0.8491 and 0.8698, respectively.

## Secondary internal validation

In the secondary internal validation set, among the three formulas, Sampson's formula exhibited significantly higher performance compared to the other two, whereas the Martin formula demonstrated the lowest performance. Moreover, all three ML models outperformed the prediction formulas, particularly in predicting LDL-C levels ≥300 mg/dL. The Friedewald formula consistently underestimated LDL-C levels, especially in cases with higher TG values. When stratifying the data based on TG levels, the predictive accuracy of both the Martin and Friedewald formulas declined notably when TG levels were ≥300 mg/dL.

For cases where TG levels exceeded 400 mg/dL, the $R^2$ values for the Martin and Friedewald formulas dropped below 0, a phenomenon that indicates the models performed worse than a simple mean-based prediction. This suggests that the traditional formulas are ill-suited to handle such high TG levels, as they fail to capture the complex relationship between triglycerides and LDL-C in this range. The negative $R^2$ reflects their inability

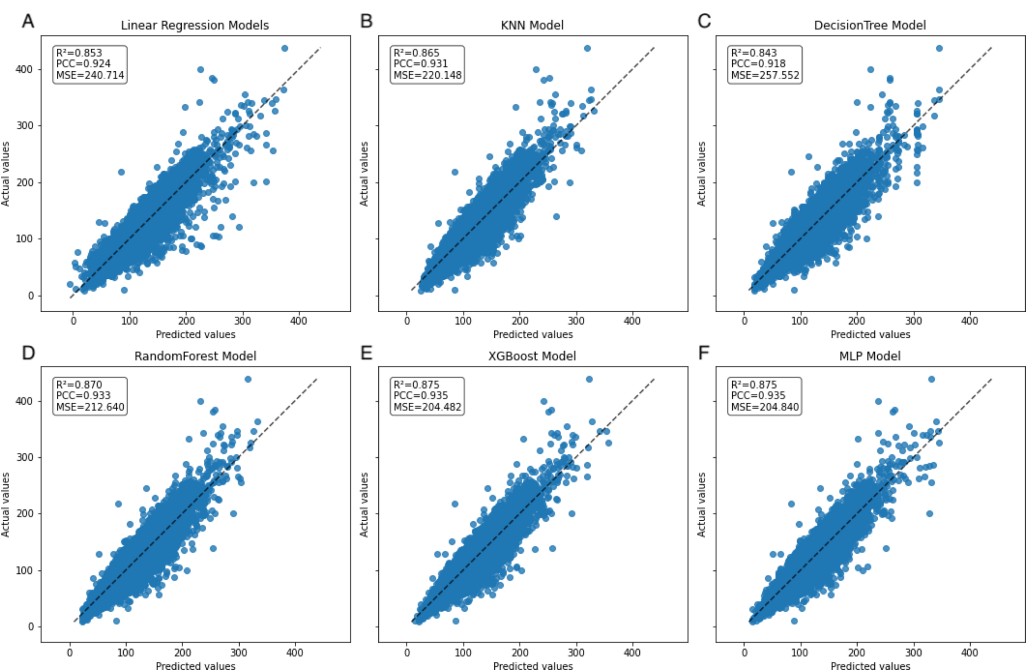

**Figure 4** **Scatter plot comparing actual and predicted values in the inner dataset.** The dashed black line represents the line of equality. Each model is labeled with its corresponding $R^2$, PCC, and MSE values in the upper left corner.

to provide reliable predictions and highlights the limitations of these formulas in hypertriglyceridemic patients.

In contrast, the machine learning models maintained strong predictive performance across all TG ranges, demonstrating greater flexibility and robustness, particularly in high TG contexts, where traditional formulas struggle (Figs. 6–9, Table 4).

## DISCUSSION

The accurate determination of LDL-C is a significant challenge in laboratory medicine; major guidelines advocate diverse dyslipidemia management strategies based on patient LDL-C levels (*Mach et al., 2020*). Due to cost limitations associated with the gold standard (*i.e.,* direct measurement of LDL-C), it is challenging to achieve widespread adoption in medical laboratories. Consequently, many medical laboratories opt for equations to estimate LDL-C values. However, the performance of these equations varies significantly and, in some instances, proves to be suboptimal when evaluated across diverse settings (*Martin et al., 2023*). This investigation represents the inaugural retrospective analysis evaluating the reliability of a ML algorithm based on hyperparameter tuning in Jilin Province. This study aims to evaluate the reliability of LDL-C estimation across different models and parameters, including variations in gender, age, TG, TC, and HDL-C. Notably, the economic challenges in Northeast China, compared to southern regions, pose additional barriers to the widespread adoption of precise LDL-C measurement methods. Nevertheless, studies indicate a higher prevalence of hyperlipidemia in the Northeast, further emphasizing

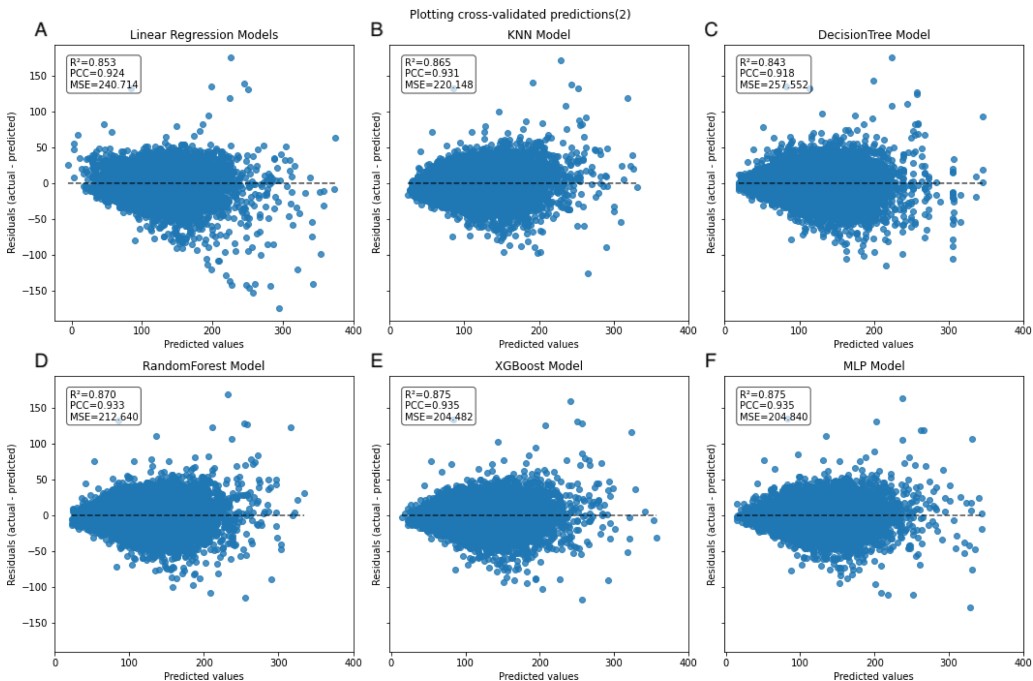

**Figure 5** **Scatter plot comparing residual and predicted values in the inner dataset.** The black dashed line represents the point where the residual difference between actual and predicted values is 0. The farther the points are from this line, the larger the absolute value of the residuals, indicating larger errors. Conversely, the closer the points are to this line, the smaller the absolute value of the residuals, indicating that the model errors are closer to 0.

the importance of accurate LDL-C measurement in this region. Additionally, the prevalence of dyslipidemia in Northeast China is 62.1%, notably exceeding the national average (*Zhang et al., 2017*). Meanwhile, the overall prevalence of the metabolic syndrome in Jilin province, Northeast China has been reported has been as high as 32.86%, which can be attributed to a genetic predisposition combined with environmental factors (*Wu et al., 2016*).

On the internal training set, the linear, KNN, XGBoost, and MLP regression models exhibit well-fitted training curves. Although the DecisionTree and RandomForest regression models exhibit less fitting compared to the former four models, they do not demonstrate complete overfitting or underfitting. Pruning these models might potentially enhance their fitting and predictive performance (*Njoku, 2019*); however, due to computational constraints, we did not perform this procedure. Although some studies propose that the KNN model can fully replace formulaic methods for LDL-C prediction (*Ghayad, Barakett-Hamadé & Sleilaty, 2022*), herein, the trained ensemble algorithm models and neural network model demonstrate superior performance.

We examined the applicability of the Friedewald, Martin and Sampson formulas in the Northeast Chinese population and assessed the performance of RandomForest regression, XGBoost regression, and MLP regression models. The results demonstrate that all ML methods outperform the closed-form equations across various performance measures,
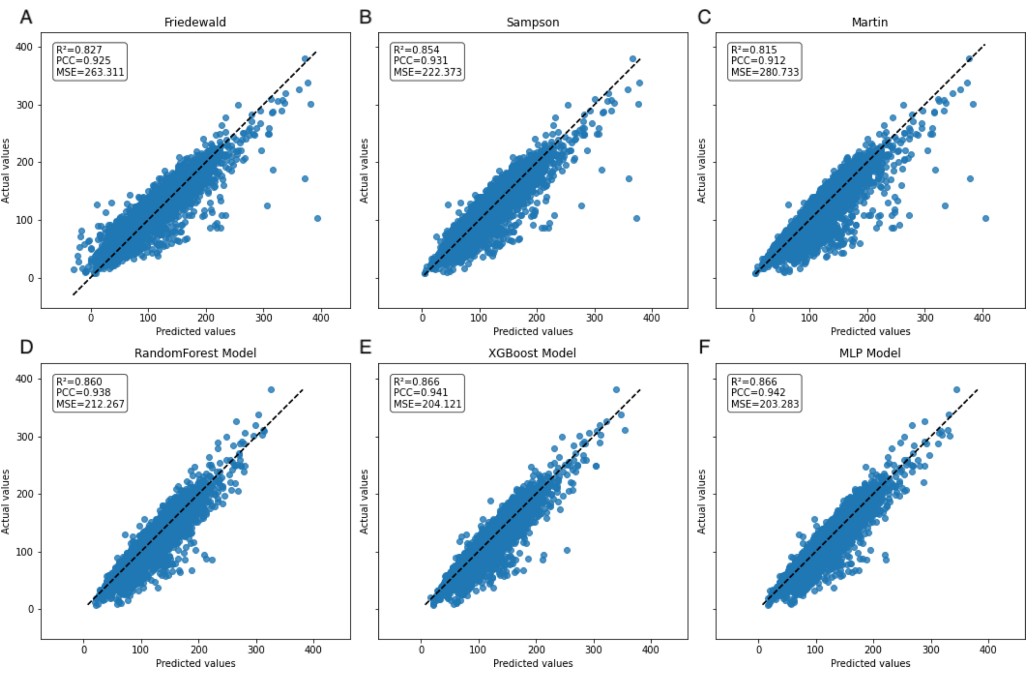

**Figure 6** **Scatter plots comparing actual *vs.* predicted values in secondary internal validation set.** The dashed black line indicates the line of equality. $R^2$, PCC, and MSE values for each model are shown in the upper left corner. Plots (A–C) represent Friedewald, Sampson, and Martin formulas, while plots (D–F) show the corresponding results for the ML models.

which is consistent with earlier research (*Çubukçu & Topcu, 2022*; *Oh et al., 2022*; *Singh et al., 2020*).

A notable constraint of the Friedewald formula is its application of a constant factor of 5 as the denominator of TG for computing VLDL-C(TG/5) (*Friedewald, Levy & Fredrickson, 1972*). In scenarios where the TG level surpasses 400 mg/dL, the ratio of TG to cholesterol in VLDL exceeds 5, primarily due to the existence of chylomicrons, their remnants, or VLDL remnants. Thus, an inflated estimation of VLDL-C occurs, which leads to an underestimation of LDL-C (*Çubukçu & Topcu, 2022*), an observation that is reflected in the data (Fig. 8A). In the secondary internal dataset, the Friedewald formula $R^2$ is even lower than 0 in the TG>400 mg/dL interval, thereby indicating that the Friedewald formula is not as effective as the benchmark model in this interval, and it is even possible that no linear relationship exists. Previous research has indicated that in the low LDL-C range, the Friedewald formula tends to underestimate values, a trend that is also reflected in the dataset (Fig. 7A) (*Barakett-Hamade et al., 2021*; *Martin et al., 2013a*). The Martin equation, which was designed to address the constraints of the Friedewald equation, occasionally exhibited superior performance compared to XGBoost, random forest, and the MLP model. However, it did not consistently outperform multiple regression or penalize regression methods across validation sets. Notably, in the same dataset, the Martin formula performed much less effectively than the Friedewald formula in predicting the interval of TG>400 mg/dL, which was evidenced by an $R^2$ value of −0.77. However, unlike the bias observed
**Table 4   Values of $R^2$ and MSE for different models at various TG intervals.**

| Model | TG category (mg/dL) | $R^2$ | MSE | PCC |
|---|---|---|---|---|
| Friedewald | >0 and <100 | 0.93 | 96.83 | 0.97 |
| | ≥100 and <150 | 0.91 | 135.65 | 0.96 |
| | ≥150 and <200 | 0.85 | 213.94 | 0.94 |
| | ≥200 and <300 | 0.73 | 380.76 | 0.90 |
| | ≥300 and <400 | 0.52 | 711.97 | 0.82 |
| | ≥400 | −0.14 | 1,498.04 | 0.68 |
| Sampson | >0 and <100 | 0.93 | 101.82 | 0.97 |
| | ≥100 and <150 | 0.91 | 129.90 | 0.96 |
| | ≥150 and <200 | 0.88 | 178.68 | 0.94 |
| | ≥200 and <300[*] | 0.79 | 305.62 | 0.90 |
| | ≥300 and <400 | 0.61 | 578.33 | 0.82 |
| | ≥400 | 0.15 | 1,118.14 | 0.67 |
| Martin | >0 and <100 | 0.93 | 94.32 | 0.97 |
| | ≥100 and <150[*] | 0.91 | 124.00 | 0.96 |
| | ≥150 and <200[*] | 0.88 | 172.72 | 0.94 |
| | ≥200 and <300 | 0.77 | 331.29 | 0.90 |
| | ≥300 and <400 | 0.46 | 805.70 | 0.82 |
| | ≥400 | −0.77 | 2,320.61 | 0.64 |
| RandomForest | >0 and <100 | 0.93 | 94.95 | 0.97 |
| | ≥100 and <150 | 0.90 | 142.89 | 0.96 |
| | ≥150 and <200 | 0.87 | 185.36 | 0.94 |
| | ≥200 and <300 | 0.78 | 321.81 | 0.91 |
| | ≥300 and <400 | 0.63 | 554.82 | 0.86 |
| | ≥400 | 0.43 | 740.54 | 0.78 |
| XGBoost | >0 and <100[*] | 0.94 | 89.25 | 0.97 |
| | ≥100 and <150 | 0.91 | 132.44 | 0.96 |
| | ≥150 and <200 | 0.88 | 174.44 | 0.95 |
| | ≥200 and <300 | 0.78 | 313.78 | 0.91 |
| | ≥300 and <400 | 0.63 | 552.45 | 0.86 |
| | ≥400 | 0.44 | 737.84 | 0.78 |
| MLP | >0 and <100 | 0.94 | 90.67 | 0.97 |
| | ≥100 and <150 | 0.91 | 136.48 | 0.96 |
| | ≥150 and <200 | 0.88 | 180.40 | 0.95 |
| | ≥200 and <300 | 0.79 | 306.50 | 0.91 |
| | ≥300 and <400[*] | 0.65 | 522.68 | 0.87 |
| | ≥400[*] | 0.45 | 719.12 | 0.78 |

**Notes.**
[*]Indicates that the model exhibits superior predictive performance in the current interval.
  All Pearson correlation coefficients obtained have $P$-values less than 0.001.

with the Friedewald equation, the Martin equation tends to overestimate in the high TG range. However, due to its excellent variation coefficient, it exhibits a more optimal prediction performance than the ML model in the 100–200 mg/dL TG interval. Compared to the instability of the previous two formulas, the Sampson formula demonstrates superior

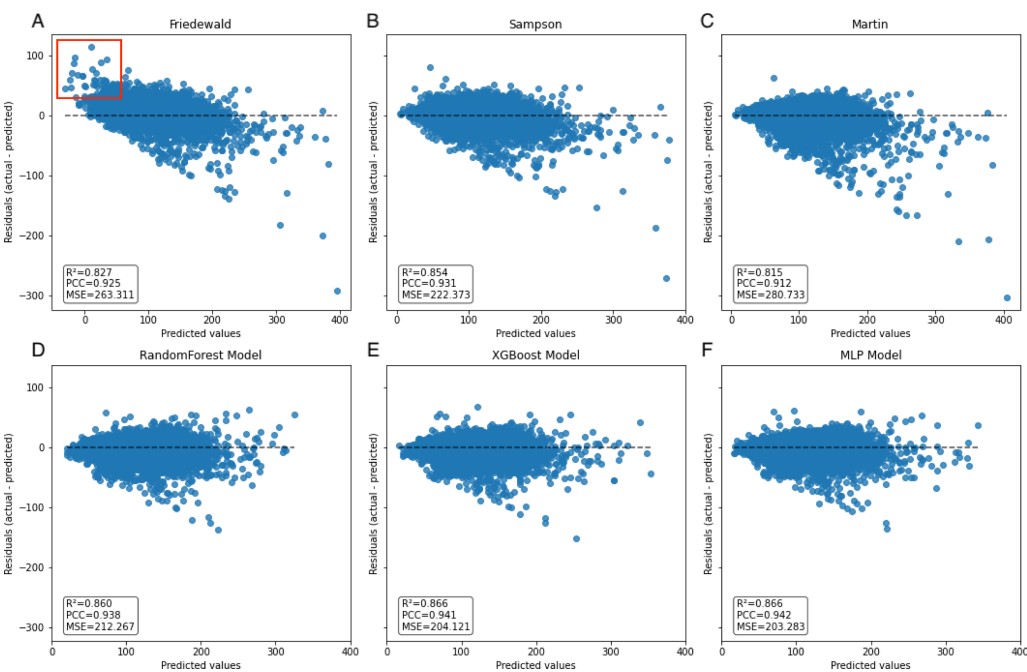

**Figure 7** **Scatter plot comparing residual and predicted values in secondary internal validation set.** The black dashed line represents the point where the residual difference between actual and predicted values is 0. The further away from this line, the greater the error between predicted and actual values. The red boxed area indicates an underestimation of the Friedewald formula in the lower LDL-C range.

predictive capability, especially in the interval where TG ranges from 200 mg/dL to less than 300 mg/dL, and exhibits more optimal predictive ability than ML models. Additionally, because the formula is closed, it offers greater convenience in calculation compared to the Martin formula. *Erturk Zararsiz et al. (2022)*'s report indicates that among the three formulas, the Sampson formula demonstrates superior predictive performance when LDL-C results are measured together with Roche, a finding that aligns with the results. *Sampson et al. (2020)* confirm that there is still a high predictive value for TG in the 400–800 mg/dL range, although this is not fully reflected in the dataset. Contrastingly, Sampson's predictive value for patients with TG >400 mg/dL is higher than that of the Friedewald and Martin formulas, and it remains lower than that of machine-learning models and cannot replace them.

Our findings demonstrate that traditional LDL-C estimation formulas, including the Friedewald equation, exhibit a significant decrease in predictive accuracy within the TG >300 mg/dL interval. This decrease in accuracy is particularly concerning given the increased cardiovascular risk associated with higher TG levels, highlighting the necessity for more reliable estimation methods in this patient subset. The results from our machine learning models, especially random forest and XGBoost, indicate superior performance in maintaining high predictive accuracy across all TG levels, including the challenging TG >300 mg/dL interval.

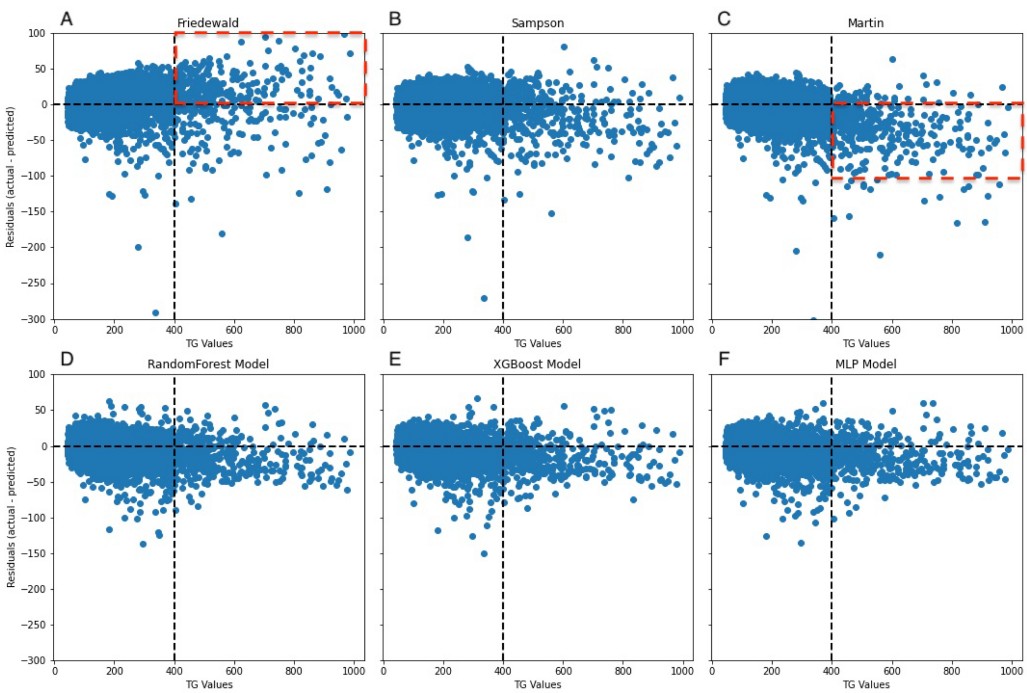

**Figure 8** **Residuals between actual and predicted values for different TG values.** The dashed line on the $x$-axis represents TG = 400 mg/dL, whereas the dashed line on the $y$-axis indicates the line where the actual value is subtracted from the predicted value of 0. In (A), the red dashed portion signifies the underestimation of the actual values by the Friedewald formula in the interval where TG $\geq$ 400 mg/dL, whereas in (C), the red dashed line signifies the overestimation of the actual values.

*Anudeep et al. (2022)*'s study found that ML models such as XGBoost and random forests model can be utilized to predict LDL-C more accurately than traditional linear regression formulas. However, in the research, these models did not consistently provide optimal predictive value, particularly in certain TG concentration ranges, such as the 100–300 mg/dl interval. Moreover, the process of hyperparameter tuning and model training incurred significant time and computational costs. Nonetheless, in the $\geq$400 mg/dl TG interval, ML models indeed exhibited superior predictive value compared to traditional linear formulas. The MLP model is typically an artificial neural network characterized by multiple layers of nodes or neurons, with each layer connected to the next (*Glorot & Bengio, 2010*) and is a versatile and widely utilized model capable of learning complex patterns and relationships in data. The MLP model demonstrates strong predictive value for complex data structures, an advantage that was not evident in the dataset. Specifically, the predictive scores of the MLP model did not significantly outperform those of the other two models. Furthermore, the sensitivity of MLP to parameters, its dependence on dataset size, and its comparatively time-consuming nature compared to the other two models limit its applicability (*Livingstone, Manallack & Tetko, 1997*). Notably, prior to training the model, we conducted feature selection, a process aimed at enhancing model efficiency, reducing computational complexity, and improving the predictive accuracy of regression
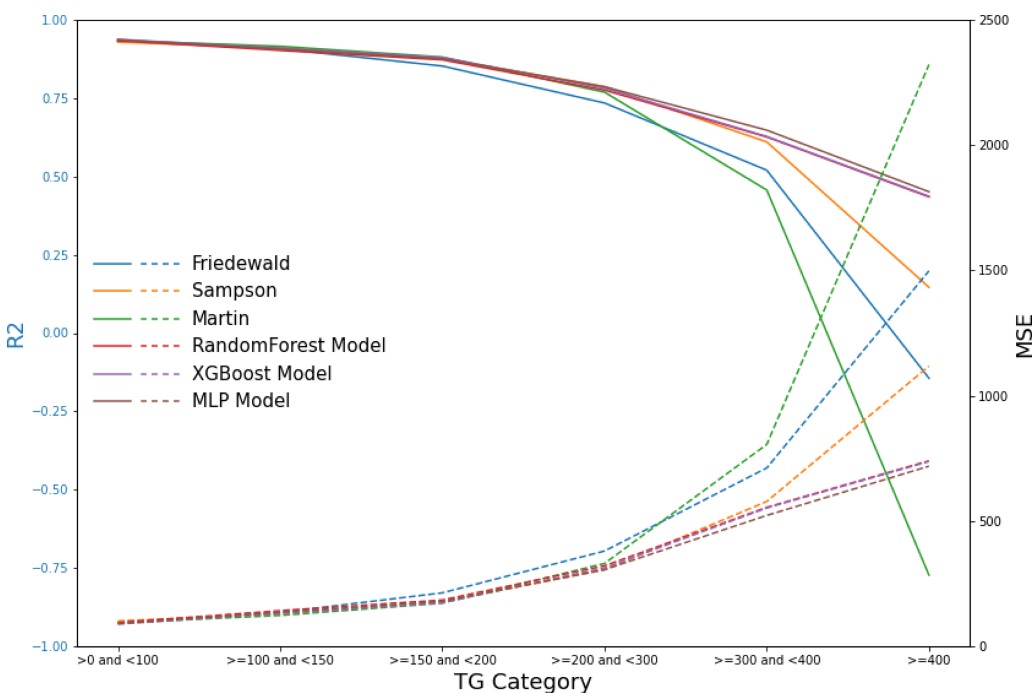

**Figure 9** Line graphs of assessment metrics across various models within distinct TG intervals. The data were stratified using TG values of 0, 100, 150, 200, 300, and 400 mg/dL as thresholds. The solid lines of different colors represent the $R^2$ of different models, and the dashed lines indicate the MSE.

tasks, as well as the generalization to unknown feature data (*Kumar & Minz, 2014*). During a similar study conducted by *Fan et al. (2022)*, it was observed that machine learning models were less susceptible to the influence of age. Instead of assessing feature importance after training the models, we employed the Mean Decrease in Impurity (MDI) method to evaluate feature weights before formal model training and fitting. This approach facilitated hyperparameter tuning, thus leading to considerable savings in computational and time costs.

Our open-source machine learning model offers a flexible solution for LDL-C estimation across various healthcare settings. Hospitals can adapt the model to their specific patient populations by retraining with local data, ensuring optimal performance. This adaptability, combined with the model's ease of integration into electronic health records, supports broad applicability and enhances cardiovascular risk assessment.

## LIMITATIONS

The current study exhibits the following limitations: (1) the absence of beta-quantification as a reference method, which could potentially enhance result accuracy, and (2) the lack of pruning operations on the decision tree and random forest models, possibly leading to an underestimation of their predictive capabilities. (3) The study evaluated only the validity of the formulas and models on the Roche platform, thereby necessitating further multicenter research on various analyzer platforms; thus, external validity was established.

(4) The lack of true external validation represents a limitation of this study, as it may affect the generalizability of the model's performance to completely independent datasets. (5) Additionally, the ML models were trained solely on data from our laboratory, making them specific to the chosen dataset. While the Friedewald and Martin formulas may exhibit broader applicability, the generalizability of our ML models for estimating LDL-C levels in diverse populations requires additional validation.

## CONCLUSION

Accurate LDL-C determination, especially in high TG ranges, remains challenging. Our study demonstrates the effectiveness of machine learning models in overcoming this challenge, showing better predictive performance compared to traditional methods like the Friedewald equation. These models could enhance cardiovascular risk assessment by providing more precise LDL-C estimates, potentially leading to more informed treatment decisions. However, integrating these models into routine clinical practice necessitates further validation and addressing practical issues like data availability and computational resources. Future research should focus on validating these models across diverse populations and evaluating their long-term predictive performance to ensure their broad applicability and reliability.

### Funding
The authors received no funding for this work.

### Competing Interests
The authors declare there are no competing interests.

### Author Contributions
- Jing-Bi Meng conceived and designed the experiments, performed the experiments, analyzed the data, prepared figures and/or tables, and approved the final draft.
- Zai-Jian An conceived and designed the experiments, authored or reviewed drafts of the article, and approved the final draft.
- Chun-Shan Jiang conceived and designed the experiments, analyzed the data, authored or reviewed drafts of the article, and approved the final draft.

### Human Ethics
The following information was supplied relating to ethical approvals (i.e., approving body and any reference numbers):

The Ethics Review Committee of Yanbian University Hospital approved the study (Ethics No. 2024665).

### Data Availability
The raw data, feature scaler used for training, the trained model, and related code are available at Zenodo: Meng, J.-B. (2025). Machine Learning-Based Prediction of LDL Cholesterol: Performance Evaluation and Validation. Zenodo. https://doi.org/10.5281/zenodo.14722724.

### Supplemental Information

Supplemental information for this article can be found online at http://dx.doi.org/10.7717/peerj.19248#supplemental-information.

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
