# Peer review of "Machine learning-based prediction of LDL cholesterol: performance evaluation and validation"

_PeerJ, doi:10.7717/peerj.19248_

## Round 0.1 · original submission · Major Revisions

Please address concerns of both reviewer and revise manuscript accordingly.

·

Basic reporting

The language used throughout the article is clear, concise, and conveys the idea effectively.

Right from the introduction, the paper focusses on the Chinese population. Stating population statistics of lipid abnormalities as well as relevance of computation of LDL-C, all were based on the China region.

The literature review is comprehensive, however, a very few recent and relevant literature addressing the same issue is missing in the paper.

The figures and tables effectively illustrate the workflow and findings, enhancing comprehension.

The table referenced in Line 169 does not correspond to the information presented.

Experimental design

The feature importance assessed using Mean Decrease Impurity (MDI) is based on training dataset. The predictive ability of features on unseen data is uncertain. Can state if that is an assumption.

The feature importance was assessed prior to ML modelling. However, after obtaining the results, did the authors analyse which feature’s contributions significantly influenced the outcome. This could help identify model’s predictions as interpretable and meaningful.

Validity of the findings

A critical assessment of the relevance and significance of the study would help validate the research findings in the state-of-the-art. This is missing since the need for computing LDL-C in this manner - is it clinically significant (though the complexity of > 300 measurements are mentioned) - this needs to be clarified, as this is the foundational motivation.

The exclusion of outliers, optimisation of hyperparameters, and validation with appropriate statistical analysis contributes to the robustness of the study. More transparency into what hyperparameters, and tuning approach were explored could support the credibility of the findings. These details need to be given as well.


An outline on how this solution approach could be effectively implemented in the real-world setting would strengthen the paper and enhance its relevance and applicability.

Conclusion is brief, and lack comprehensive insights. A more expanded conclusion could better summarize the key findings enhancing the overall understanding of the paper.

Reviewer 2 ·

Basic reporting

No comment

Experimental design

See attached review

Validity of the findings

No comment

Additional comments

See attached review

Annotated reviews are not available for download in order to protect the identity of reviewers who chose to remain anonymous.

---

## Round 0.2 · accepted · Accept

All issues pointed by the reviewers were adequately addressed and the revised manuscript is acceptable now.